# Environmental Sustainability Perspectives of the Nordic Diet

**DOI:** 10.3390/nu11092248

**Published:** 2019-09-18

**Authors:** Helle Margrete Meltzer, Anne Lise Brantsæter, Ellen Trolle, Hanna Eneroth, Mikael Fogelholm, Trond Arild Ydersbond, Bryndis Eva Birgisdottir

**Affiliations:** 1Division of Infectious Diseases and Environmental Health, Norwegian Institute of Public Health, 0213 Oslo, Norway; 2National Food Institute, Technical University of Denmark, Kgs. 2800 Lyngby, Denmark; 3Swedish Food Agency, 751 26 Uppsala, Sweden; 4Department of Food and Nutrition, University of Helsinki, 00014 Helsinki, Finland; 5Statistics Norway, 0131 Oslo, Norway; 6Unit for Nutrition Research, Faculty of Food Science and Nutrition, University of Iceland and Landspitali-University Hospital, 101 Reykjavik, Iceland

**Keywords:** environment, sustainability, Nordic diet, FBDGs, food systems

## Abstract

“The Nordic diet” is an umbrella term that encompasses any interpretation that combines Food-Based Dietary Guidelines (FBDGs) with local Nordic foods. The five Nordic countries have collaborated on Nordic Nutrition Recommendations for forty years, including FBDGs, so their national guidelines are similar. The countries also share similar public health issues, including widespread nonconformity to the guidelines, although in different ways. The aim of this concept paper is to discuss environmental sustainability aspects of the Nordic diet, describe the status of and make suggestions for the inclusion of sustainability in future work on the Nordic diet. We exploit the sustainability–health synergy. A food intake more in line with the current FBDGs, which emphasises more plant-based and less animal-based foods, is necessary for high environmental sustainability. In turn, sustainability is an important motivator for health-promoting dietary shifts. Policy development requires long-term efforts. Since the Nordic diet can be considered a further development and improvement of old, traditional diets, there is huge potential to formulate a Nordic diet that benefits both human and planetary health. It is time for concerted engagement and actions—a new Nordic nutrition transition.

## 1. Introduction

The United Nations (UN) have defined 17 Sustainable Development Goals (SDGs) for the year 2030. Most of these goals have diet and health as a prerequisite for achievement [1]: the main aspects are related to removing hunger, securing enough safe and nutritious food for all, decreasing the prevalence of non-communicable diseases (NCDs), reducing social inequalities and securing environmental sustainability, including combating climate change and its impacts. In the wake of the 2015 SDG agreement, a wealth of literature has emerged where diet, health and environmental sustainability are studied in combination [2,3,4]. Their conclusions are unanimous: energy balance and varied, more plant-based diets contribute to achieving the SDGs.

The five Nordic countries, Denmark, Finland, Iceland, Norway and Sweden, also referred to as “the Nordics”, share many common values, illustrated in their ability to combine a generous tax-funded welfare system with efficient public administration and a competitive business sector [5]. Furthermore, the Nordic countries are among the highest ranked in international comparisons on health, welfare and well-being, the latter clearly demonstrated through the World Happiness Index produced by the United Nations Sustainable Development Solutions Network. For six happiness indicators, social support, generosity, healthy life expectancy, perception of corruption, gross domestic product per capita and freedom to make life choices, the Nordic countries were again at the very top in 2019, with Finland in first place, followed by Denmark (2nd), Norway (3rd), Iceland (4th) and Sweden (7th) out of the 156 countries included [6].

For over 60 years, the Nordic countries have had a unique regional collaboration within a number of societal areas, including politics, economics and culture. Since 1980, they have collaborated on Nordic Nutrition Recommendations. Furthermore, the Nordic countries have committed strongly to the SDGs and recently agreed to extend their climate and environment efforts to create a climate-neutral and sustainable Nordic area according to the aims of the Paris agreement and Agenda 2030 [7].

Within the Nordic countries, there is a similar prevalence of overweight and obesity, malnutrition and dietary-related non-communicable diseases (NCDs) such as coronary heart disease, certain cancers (e.g., colorectal and breast) and type 2 diabetes [8,9]. Many of the common Nordic health policy priorities are diet related, including a reduction in social health inequalities, NCD prevention and promotion of healthy ageing. Each Nordic country has distinct dietary characteristics but they share certain key features such as higher potato, milk, coffee, sugar-sweetened beverages and fish consumption, and lower fruit and vegetable consumption than most other European countries [10].

The long-term collaboration on the Nordic Nutrition Recommendations (NNRs) has contributed to a common knowledge base. The latest version (NNR5) is based on systematic reviews by Nordic experts in addition to research and reviews from the individual countries and recent international literature [11]. It includes a chapter on Food-Based Dietary Guidelines (FBDGs), giving the current best evidence for dietary components that are most likely to fulfil nutritional needs, extend life and prevent NCDs, but that are also compatible with the traditional Nordic diet (Table 1). With the NNRs as a backbone, the individual, country-specific FBDGs are very similar between the Nordic countries, although minor differences are found that encompass local conditions, for example, in culinary traditions and fortification policy.

The NNR5 FBDGs focus on plant-based foods. They also recommend low-fat dairy products and fish, and recommend limiting the consumption of processed meat, red meat, added sugar, salt and alcohol. Environmental sustainability is discussed in one chapter in NNR5, with an estimate that changes in the current Nordic diet towards the FBDGs would result in substantial changes towards lowering greenhouse gas (GHG) emissions [11]. However, the degree to which sustainability is included in current national FBDGs differs between the Nordic countries. Sweden was internationally recognised for incorporating sustainability aspects to each food group already at the launch of their FBDGs in 2008, with an update in 2015. Iceland included comments on waste, meat and greenhouse gases, while Norway and Finland included a separate chapter on environmental aspects of the diet in their national dietary recommendations in 2011 and 2014, respectively. In Denmark, a separate chapter on climatic aspects of the diet was included in the evidence base for the Danish guidelines for diet and physical activity (2013), including a suggestion to supplement advice for each food group of the Danish FBDGs [12,13]. Unlike Sweden, these aspects were not integrated in the separate food group recommendations in Denmark, Finland, Iceland and Norway.

The “New Nordic Diet” (NND) concept emerged in 2004 as guidelines for Nordic chefs to explore Nordic-grown produce [14] and as dietary recommendations for the general public to increase focus on locally grown food [15]. The NND concept is not science-based like the FBDGs and may be considered as a more heuristic concept. It is based on four core principles: health, gastronomic potential, sustainability and Nordic identity. The Chefs Manifesto, adopted by the Nordic Council of Ministers as the ideology of the New Nordic Food programme, further emphasises wild, foraged, local, fresh and highly palatable cuisine [14]. The NND concept has partly penetrated mainstream Nordic consciousness and has captured global media attention [15]. Accordingly, dietary patterns adhering to the Nordic FBDGs with additional focus on local food, have generally been dubbed “The Nordic diet” in studies relating to health [16]. “The Nordic diet” can thus be considered an umbrella term that encompasses any interpretation that combines FBDGs with local, Nordic foods, and this definition will be used in this article.

The aim of this concept paper is to discuss environmental sustainability aspects of the Nordic diet, starting with the current state of the art and later suggesting how to encompass the SDGs in future work on the Nordic diet.

## 2. Dietary Guidelines and Food Consumption

### 2.1. Food-Based Dietary Guidelines in the Nordic Countries

The NNRs are evidence-based recommendations based on research, mainly by Nordic universities and governmental research institutions [11]. The recommendations have been revised and updated five times over the last forty years, most recently in 2012 (NNR5), with a sixth update planned for 2022. They have evolved from an initial focus on dietary reference values for different age groups, fulfilling the energy, macro- and micronutrient needs of the Nordic populations, towards looking more at the entire diet with all its components, giving simple food group-based dietary guidelines for children from 2 years and onwards. Since 2004, they have also included physical activity. The NNRs are intended to promote optimal nutrition as a basis for good health to the general population, addressing the most important current issues. In the latest edition, the role of dietary patterns associated with a reduced risk of diet-related NCDs and maintaining energy balance (i.e., energy intake equal to energy expenditure for a healthy body weight) were emphasised (Table 1). The NNRs and any participating experts have no conflicts of interest and the food industry has never been invited to participate in the process.

Typically, the dietary recommendations are used as guidelines when planning diets, especially in the public sector for people of different age groups. They have also been used to formulate nutritional policies, e.g., vitamin D fortification in Finland, Norway and Sweden. Furthermore, they are extensively used to evaluate dietary intake in the population and for nutrition education and information, as well as health promotion.

### 2.2. The Nordic Diet

Traditionally, the Nordic FBDGs have focused on food and food groups available in the Nordic countries, with little consideration of what could be locally produced (Table 2). However, with increased environmental awareness and networks such as the New Nordic Food project supported by developments in local supply chains, there is a growing interest in local food as part of future FBDGs. The NND program [17] initiated a number of activities, projects and campaigns to increase knowledge, communication, awareness and visibility of local foods to reinforce their place in the Nordic and global food markets. It reinforced the concept of Nordic Food, with communication as a key tool.

The Nordic climate restricts crop diversity, although plant foods such as berries, apples, pears, root vegetables, cabbage, cauliflower, curly kale, onions and mushrooms, as well as barley, wheat, spelt, oats, buckwheat and rye thrive in these countries. The same applies to rapeseed for canola oil production, linseed and hazelnut bushes (Table 2). During the summer months, local greenhouse produce such as tomatoes, peppers and cucumbers extend the range. Lean and oily fish from local lakes and seafood along the coastlines are widely available, both wild and farmed. Milk (mainly from cows but also sheep and goat for cheese) and meat from both wild and domesticated animals (beef, lamb, pork and game) and birds (farmed and wild birds, plus their eggs) are available. Foods found in nature, e.g., wild berries, mushrooms, herbs such as thyme and oregano, and marine products such as kelp, are attracting growing interest among the Nordic populations. Traditionally, legumes such as peas and some bean varieties have been grown in the Nordic countries, with the latter being mostly suitable for animal fodder if grown north of latitude 60° N. In Sweden, land allocated for bean production—also for human use—has increased substantially over the last decade [18] and there are ongoing projects to extend the range of products, e.g., to green soybeans. There is a potential to grow an array of new types of plant foods in the Nordic countries with increased demands and policy change. Climate change with higher temperatures helps, but suitable varieties are needed, as seen in Norwegian wheat production—in good years, the country is essentially self-sufficient with food-quality wheat, previously unthinkable so far north. Greenhouse-produced foods also have a large and growing potential; new facilities are mostly heated with renewable energy, like geothermal heat (Iceland), heat pumps driven by “green” electricity, and excess heat from industrial processes or biofuels/biogas.

### 2.3. The Nordic Diet and Health

Both Nordic and local funding programmes have supported extensive studies, some multicentre, on traditional Nordic foods and diets based on the Food-Based Dietary Guidelines. [19] These studies have added evidence about any positive health effects. One example is the OPUS study (Optimal well-being, development and health for Danish children through a healthy New Nordic Diet), a crossover intervention school meal trial among 834 children aged 8–11 years [19]. Although the exact definition of the Nordic diet may vary among studies, the versions are comparable, the main difference being in the grade of local or seasonal food emphasised. The projects have resulted in a number of scientific publications. Intervention studies with endpoints such as cardiovascular risk factors (blood lipid profile, insulin sensitivity and blood pressure) [20,21,22,23] and low-grade inflammation [24] have shown beneficial effects with a designed Nordic diet. Observational cohort studies have also shown that higher adherence to the Nordic diet is associated with lower total mortality [25], reduced risk of colorectal cancer [26], less body fat and healthier weight development [27], and reduced obesity-related markers of inflammation [28]. Higher adherence to the Nordic diet has also been associated with more favourable pregnancy outcomes, e.g., optimal weight gain during pregnancy and improved foetal growth [29], lower risk of preeclampsia and spontaneous preterm delivery [30], and long-term maternal weight development [31]. Furthermore, in a recent publication from the World Health Organization (WHO), the Nordic diet is evaluated as similar to the Mediterranean diet regarding health-promoting properties [16].

### 2.4. Current Food Consumption

Each Nordic country conducts dietary surveys to monitor the eating habits of their populations. Despite a number of very positive changes compared to dietary patterns approximately 50 years ago, there is still room for improvement to better align with current Nordic FBDGs (Table 1). The most recently published reports from the dietary surveys of adults are from the years 2010–2017 [32,33,34,35]. Generally, while each Nordic country has distinct dietary patterns, the region shares some positive trends and faces similar challenges. Progress has been made in several domains. For example, vegetable consumption has increased in all the Nordic countries and fruit consumption has increased [36] except in Denmark, where the intake of whole grain has increased [37]. In 2011, a simple, common Nordic lifestyle monitoring system was developed, with a follow-up in 2014, using the same questionnaires in all the Nordic countries [9]. This enabled direct comparisons about dietary intake, physical activity, and overweight and obesity. The collection of baseline data in 2011 showed similarities and differences between the countries. From the data, it has been estimated that in the Nordic region, when scored with a healthy diet index, one in every five adults has an unhealthy diet, with a higher proportion among men than among women (24.7% vs. 18.4%), and every seventh child, with a higher proportion among boys than girls (17.5% vs. 13.7%). Furthermore, between 2011 and 2014, the intake of fish and wholegrain bread and the proportion with a high consumption of sugar-rich foods decreased whereas the proportion with a high consumption of foods rich in saturated fat increased. The social inequalities in dietary habits were large [9]. In common with other high- and middle-income countries, unhealthy diets with high intake of ultra-processed, energy-dense nutrient-poor foods [38,39], and diet-related metabolic conditions, are leading risk factors for NCDs in the Nordic countries [40], with a considerable social gradient.

## 3. Sustainability Aspects of FBDGs in the Nordic Countries

In the annual UN report on performance according to the 17 SDGs, the Nordic countries seem to be doing very well, with Denmark, Sweden and Finland ranking 1st, 2nd and 3rd respectively, Norway 8th and Iceland 14th [41]. However, they also face major challenges in implementing many of the SDGs. The report states that no country is on track for achieving all 17 goals, with major performance gaps even among the top countries on SDG 12 (Responsible Consumption and Production), SDG 13 (Climate Action), SDG 14 (Life Below Water) and SDG 15 (Life on Land). The UN Food and Agriculture Organization (FAO) states that food and agriculture are at the very heart of the 2030 Agenda and are linked to each of the 17 goals, meaning that food system interventions could yield co-benefits across the SDGs [1]. Given today’s challenges, where global food production threatens climate and ecosystem stability and is one of the largest drivers of environmental degradation [42,43], FBDGs need to take into account both human health and environmental sustainability. This is highlighted in two recent publications. The EAT-Lancet report, launched in January 2019, has a global, healthy, flexitarian reference diet as a starting point, given as suggested amounts (as mean + range) consumed from different food groups. Some ranges are very wide, giving flexibility to adjust the recommendations to local conditions. The recommendations are made according to the impact such a diet would have on life expectancy and NCDs, and then evaluated for climate and natural resources impact, showing a substantially reduced environmental footprint of food production and consumption if the global averages they suggest are reached [4]. However, the report strongly emphasises the need to scale the average amounts given to local production conditions, resources and culture. This was investigated in a second report, a thorough analysis of Nordic diets in light of the EAT-Lancet report [36], with the main focus on Sweden, Norway, Denmark, and Finland. It clearly shows that the current Nordic diets place pressure on the environment, both domestically and abroad. The report makes it clear that dietary pattern changes in the direction of current Nordic FBDGs or EAT-Lancet report recommendations with even less meat and milk will require large structural changes at many levels in the Nordic food systems. The EAT-Lancet global reference diet may serve as one of the inputs for future FBDGs, and an update of the Nordic FBDGs should be based on a systematic review of the health-based evidence behind the report, as well as other new, relevant evidence, considered in a Nordic context. For example, the amounts suggested for red meat, legumes, potatoes and added sugar differ substantially from the current FBDGs in the Nordic countries.

On the production side, the Nordic food system is fairly well integrated in a global context. There is extensive geographical specialisation, and Nordic food production may generally be considered efficient [44]. Most of the important crops produce yields at or above the European average, and in most development work, environmental and sustainability criteria are emphasised alongside yield. Most imports are designed to complement and improve the efficiency of current local production, e.g., protein fodder components to make better use of local resources [45,46]. Long-term goals include improved efficiency in local production of legumes and oil seeds, which are important fodder ingredients but are also for human consumption. This may reduce the global footprint of the Nordic diet, as well as improve sustainability through increased agricultural diversity (including improved Nitrogen fixation).

A recent report from the World Resources Institute shows how a combination of efforts both within food production sectors and within food consumption may contribute to reduce GHG emissions by 2050 globally, as indicated in Figure 1. To achieve the goals, (1) changes in food production (transformation), (2) a reduction in food loss and waste, and (3) changes in the food composition of diets are needed, illustrating that the contribution to environmental impact reduction from dietary and food system changes may be substantial [2].

There are also estimates of the impact on other environmental aspects. For example, a study by Blackstone et al. from 2018 shows that the dietary contribution to the impact on climate change, land use, water depletion, freshwater eutrophication, marine eutrophication and particulate pollution varies strongly for different food groups [47].

It is important to ensure that future recommendations reflect climate and other environmental aspects affected by dietary intake. So far, there is a lack of well-defined and quality assured metrics and a common approach to assessing the multiple components of diet sustainability. This has impeded progress in generating the evidence needed to ensure the quality and applicability of new guidelines. Various indicators have been used to measure the environmental impact of diets. In a review paper that counted which indicators were most commonly used when estimating the GHGs of foods through various stages of production, use, and recycling, the Life Cycle Assessment (LCA) approach was the most frequently applied method. However, this was followed by measurements of land use and estimates of meat and dairy consumption, then diet quality, management practices such as organics, and water use. Local or seasonal food procurement, cost of diets and revenue generation, eutrophication potential and health parameters came lower on the list, while only a few publications had included factors such as animal welfare, fisheries or biodiversity aspects [48].

Several problems are implied in applying general methods, e.g., water use may be an important factor globally, but tends to be less relevant in most Nordic contexts. The same applies to land use, where long-term effects and alternative uses are decisive for the assessments. Another effect to remember is that a slight fall in methane emissions from established dairy and meat production will not contribute to further warming, although an increase in methane emissions will contribute substantially [49].

This concept paper focuses on the Nordic countries’ food systems situation according to the following important aspects: GHG emission, land use, water use and biodiversity, and the impact of local and organic production and food waste. Also, a more comprehensive analysis of ecosystem stability, as suggested in the paper by Gustafson et al. [50], and human and environmental toxicity, should be considered in future NNR work. Such aspects are outside the scope of this paper.

### 3.1. Climate Impact—GHG Emissions (GHGs) of a Nordic Diet

Substantial research already exists showing the impact on GHG emissions if current diets change towards the pattern recommended in the FBDGs, and of different dietary patterns such as vegetarian or vegan diets. In Denmark, only small reductions in estimated GHG emissions, approximately 2–4%, were found when evaluating three habitual Danish dietary patterns (the Traditional, the Green and a Fast food pattern) against the FBDGs and Nordic Nutrient Recommendations, focusing on making as few and small changes as possible. These calculations were based on limiting the weekly intake of red meat (and processed meat) to a maximum of 500 g, while the excess of red and processed meat was substituted with white meat (chicken), which has no limitations according to the Danish FBDGs. The total amount of meat was not changed. Other changes were also in accordance with the main Danish FBDGs as described above. In addition, the diets were then further modified, substituting foods with high GHGs with foods with lower GHGs within each food group. The estimated change became a 22–29% GHG reduction. Evaluating a vegetarian diet from the Green food pattern (including milk and egg products) and fulfilling the Nordic Nutrition Recommendations (NNR5) showed a 20% reduction [51]. This is in line with the results of a systematic Swedish review showing reductions of 0–35% by changes to healthy diets and 20–35% and 25–55% for vegetarian and vegan diets, respectively [52]. Due to all the complexities and uncertainties involved, these numbers should be considered as illustrations rather than precise estimates. In addition to the methane problem mentioned earlier, several factors with large uncertainties, like nitrous oxide emissions and changes in soil organic matter are important for the overall picture. According to whole-diet estimations, a vegan diet seems to have the lowest carbon footprint (GHG emissions). However, other diets with mostly plant-based foods (e.g., traditional Mediterranean diet, lacto-vegetarian diet, etc.) are also much lower in carbon footprint compared to current dietary habits in most high and middle-income countries [3]. The main component in diets with a reduced carbon footprint seems to be a lower intake of red meat, particularly beef, replaced by fish or plant-based protein sources [3,53]. In the development of the healthy and sustainable FBDGs in the Netherlands, optimisation models were made for different diets, finding that dietary requirements for health could be reached without eliminating meat or dairy products [54]. This supports the EAT-Lancet report, which identified the plant-based flexitarian diet as optimised for health while meeting the sustainability criteria [4]. The Nordic diets include foods which were not included in the EAT-Lancet Commission analyses, such as coffee, tea and alcoholic beverages. As these foods might contribute considerably to the environmental impact of the diets [36], they must be included in comprehensive assessments. In Denmark, it has been estimated that coffee, tea and alcoholic beverages account for approximately 14% of the GHG emissions of an average diet (10 MJ) [51].

Bryngelsson et al. (2016) studied the GHG emissions in theoretical diets with reduced meat consumption, using scenarios for current and developed technologies. Both reduced meat consumption and improved technologies contributed to lower environmental impact. Therefore, a reduction in the food-related carbon footprint can be achieved in two ways: (1) change in production/delivery technologies; (2) change in food consumption [53]. In practice, both ways are needed. However, it must be kept in mind that foods that score low on GHG emission are not necessarily healthy, such as refined sugar, and that healthy foods may not have a low carbon footprint, e.g., fruit and vegetables flown from warmer climates or domestic produce from fossil fuel-heated greenhouses [53].

### 3.2. Land and Water Use

Table 3 gives an overview of hectares of arable land in the five Nordic countries. Denmark is the only country with a large per cent of arable land, while the other four countries have less than 7 per cent. In Iceland and Norway, the situation is challenging, with most of the arable land best suited for ley and fodder production. This implies that Iceland and Norway, if they are to optimise food production based on the use of local resources, are especially suited for milk and meat production.

All five Nordic countries have high levels of food imports, resulting in substantial environmental impact outside the Nordic countries, as illustrated in Figure 2, which shows their toll on water and GHGs outside the region, including imports of fodder [36].

In normal years, the Nordic countries have high average precipitation levels, and because of increasing temperatures, they are normally listed among countries that might expect increased yields in the century to come [1]. In 2018, however, Northern Europe, including the Nordic countries, experienced extreme drought followed by a substantial reduction in harvests. In the wake of such an unexpected season, the Nordic countries joined forces to develop new water and production methods to be able to meet future extreme weather events [56].

### 3.3. Biodiversity

Agricultural biodiversity is a critical component of a sustainable food system. Without agricultural biodiversity, a food system is not sustainable [57]. Biodiversity is encompassed specifically in SDG 15: “*Protect, restore and promote sustainable use of terrestrial ecosystems, sustainably manage forests, combat desertification, and halt and reverse land degradation and halt biodiversity loss*”.

For food and agriculture, biodiversity includes the domesticated plants and animals raised in crop, livestock, forest and aquaculture systems, harvested forest and aquatic species and the wild relatives of domesticated species, plus other wild species harvested for food and other products. “Associated biodiversity” is the vast range of organisms that live in and around food and agricultural production systems, sustaining them and contributing to their output [58].

The recent Global Assessment Report on Biodiversity and Ecosystem Services draws a grim picture of the state of biodiversity [59], stating that biodiversity loss is just as catastrophic as climate change. Little is known about the current environmental impact of biodiversity loss in the Nordic countries, but like most other countries, there has been a gross reduction in varieties of grown crops and domesticated animals.

While agriculture is mostly a big driver of biodiversity loss through land use changes, e.g., deforestation, pollution and climate change, there are also some positive effects. Several red-listed species depend upon open agricultural landscapes and grazing animals that maintain semi-natural pastures and help preserve biodiversity [58].

### 3.4. Locally Produced Food

The primary food production sectors of each Nordic country vary considerably in terms of the level and type of crop production, livestock production and export orientation. As such, the environmental impact of agriculture and aquaculture varies across the five countries. Denmark is, for example, a major exporter of live animals and meat, mainly pork but also butter. Norway is the world’s largest salmon exporter, Iceland exports seafood and Sweden exports fish, while dairy is the biggest food export of Finland [60]. Although the focus on locally produced foods started with the NND projects, the interest of consumers has increased, partly due to increased awareness of climate and environmental change impacts and an increasing focus on food security (i.e., sufficient, safe and healthy food for everyone). Furthermore, assessments of the impact of different possible food systems using a range of scenarios support more focus on local food [61]. The results of this and similar analyses indicate that balancing import and export in the Nordic countries, subject to the minimisation of total environmental impact, may be a basic principle in future Nordic food policy.

Climate will always restrict Nordic crop diversity but an increasing variety of foods from all food groups can be grown or produced efficiently in the Nordic countries (Table 2) subject to the design and calibration of supporting policies [18].

### 3.5. Organic Farming

Organic farming is an agricultural system that uses ecologically-based pest control and biological fertilisers derived largely from animal and plant waste and nitrogen-fixing cover crops. All use of genetically modified products is prohibited. Compared to conventional agriculture, organic farming uses fewer pesticides, may increase soil organic matter and in some cases reduces soil erosion, decreases nitrate and phosphate leaching into groundwater and surface water, and recycles plant and animal waste efficiently [62]. These benefits are counteracted by higher food costs for consumers and generally lower yields. Organic crop yields have been found to be approximately 20–25 per cent lower overall than conventional crops, although this can vary considerably depending upon farming methods and the type of crop [63]. In many cases, organic production requires a larger area to produce the same amount of food. However, this difference becomes smaller when “best agricultural practice” methods are used to address some of the problems associated with modern, conventional farming methods [64].

If current sustainability-based dietary recommendations are followed, organic production will probably not pose a challenge to Nordic food supply. Reduced meat consumption will reduce the amount of animal fodder required, rendering organic crop levels adequate. Even with the current diet, reducing food waste alone could make organic production sufficient [61,65,66,67,68].

It should be noted that organic production is often not superior to conventional production in comparison studies, and for some parameters it may be worse. However, some agricultural system aspects support the gradual current increase in organic farming. Firstly, fertiliser and pesticides are now so expensive that organic production, with lower yields, cheaper production and somewhat higher product prices, may become economically attractive. Secondly, many of the current agricultural challenges are caused by monocultures, rough soil treatment and an almost total dominance of annuals. Furthermore, the reliance on herbicides, fungicides and insecticides makes the use of regenerative processes and ecological feedback mechanisms more difficult. Finally, chemical fertilisers tend to create pollution problems over time, e.g., with local eutrophication and water contamination. The net result is more problems with making the agricultural areas robust and functional ecological systems [69]. Today, the Nordic countries have very variable rates of organic area, production and sales, as illustrated in Table 3. If the Nordic countries are seen as one production system, an intensified organic production could sustainably feed a larger population than today. On the other hand, if meat consumption continues to increase, it will probably be difficult to be self-sufficient even with conventional production [61].

All Nordic countries have an aim to increase the share of organically produced foods within the next decades. For example, both Sweden and Finland aim to increase the share of organic food in public institutions considerably over the coming years and Denmark aims to double the organically cultivated area from 2007–2020 [18]. The Danish government has made considerable efforts to promote organic food through the popular Organic Cuisine Label, awarded to any professional kitchen serving at least 30% organic food, and awarding bronze, silver or gold depending on levels [18,70]. Recent systematic literature reviews have indicated compositional differences between organic and conventional foods and there is some evidence for potential benefits of organic food consumption from human cohort studies [64]. However, the documented beneficial health effects of eating vegetables, fruit and other foods recommended in a balanced diet are independent of production systems and there is insufficient evidence to conclude whether organic foods confer additional health benefits [64].

### 3.6. Food Waste

Taking action on diminishing food waste has been considered the easiest yet highly effective action for countries to take to decrease GHG emissions [71], so it has been high on the political agenda in the Nordic countries. For example, it has been estimated that up to 20% of food bought by the average consumer is thrown away [18]. At the Nordic level, a target of halving the amount of food waste by 2030 has been adopted [72], with concentrated media attention on the subject.

Campaigns from advisory and environmental groups and a joint Nordic campaign between the Ministries for Environment [18] have moved food waste up the national agenda. In the Nordic countries, 25 to just over 30% of municipal waste is recycled (Table 3), a percentage that the countries aim to increase over the next years. However, with every round of recycling, toxins and pollutants may also be recycled. On a regulatory note, some EU countries have banned organic landfill disposal. Alternatives to landfill include composting food waste to produce soil and fertiliser, animal feed, or used to produce energy or fuel. On the basis of food waste, food and agro-waste have also been adopted as sources for new drug leads or important phytochemicals with different therapeutic benefits in many countries, with research funding invested into the development [73].

## 4. Towards a Healthy and Environmentally Sustainable Nordic Diet

### 4.1. A Food Systems Approach

Both the EAT-Lancet report [4] and the report from the Stockholm Resilience Centre focusing on the Nordic countries [36] highlight the need for an integrated food systems approach to reach the SDGs, taking into account all steps in agricultural and aquatic production, trade, manufacturing, retailing and consumption. This will involve many sectors and should have a focus of long-term achievements and development of economically, politically, technically and environmentally robust systems. Citizens can make an impact through food choices and how they vote in government elections, while politicians may have to make unpopular decisions when introducing policies to promote sustainability and better health. A lack of majority support has never been a valid reason for not developing the measures needed.

It is important to identify and exploit possible win–win situations and develop effective strategies for change. What works in some situations and for some groups may not work for others In addition, the audience will vary. Some food system changes will mostly affect farmers, some have their main impact on industry and distribution, and some are more consumer oriented.

The basic synergy to exploit is that between health and sustainability. Health-motivated changes towards a more plant-based diet may contribute essentially to improving the sustainability of the food system, since many of the current problems relate directly to the high demand for meat, eggs and dairy products. Conversely, concern for the sustainability of food production may motivate a transition towards a more plant-based diet for individuals who have no immediate health concerns about their dietary patterns.

A broad approach focusing on the determinants of food choices in the most general sense, and food consumption patterns might be useful. This is illustrated in the social ecological model (Figure 3) that shows the multiple levels of society. The model includes the international, national and provincial public policy level (policies and subsidies, taxation, advertisement and marketing regulations, food safety, nutrition labelling and food claims, urban planning, food system and supply chain regulation); the community and organisational level (food environment, workplaces and educational settings, retail, service, community and recreational facilities); interpersonal (family, parenting and personal relationships, social networks and peer pressure/support); and the individual level (preferences, knowledge, skills, motivation, attitudes, self-efficacy and self-confidence) as well as targeting age and gender [74]. For best results, these levels should be addressed simultaneously, as active tools for change, to make fast progress to a better future.

Campaigns for health in every policy have matured in the Nordic countries over the last decade and by adding environmental aspects, they will have even more impact. Some have proven to be cost-effective, like the Danish six-a-day campaign. However, the easiest way to make major changes with sufficient impact so far seems to be through policy change, rules and regulations such as taxation, but only if (a) the tax is high enough; (b) there is a good alternative cheaper product. A combination of several methods also seems to be better than one (non-fiscal) alone [75,76]. The taxation of red meat has also been estimated in modelling exercises [77]. However, taxation may also have considerable political costs, for example if there are reactions because of perceived violations of “individual freedom” or if health becomes a competing issue with job losses (due to lower demand).

To achieve lasting dietary changes, it is often helpful to address public knowledge with a focus on food literacy and be able to communicate the importance and urgency efficiently. Recent studies indicate that the people in the Nordic countries are concerned about the environment [78]. They want regular information from trusted, transparent and balanced sources but also need extra help in their daily lives to make beneficial changes and assess their contributions. An efficient way is to create infrastructures that make healthy and sustainable choices easy.

Strong institutions and a tradition for proactive governance means that ensuring the accessibility and availability of healthy food for all in the Nordic societies may be less of an effort [79]. The Nordic countries have a vision to join the group of leading nations in sustainability and health, as demonstrated by initiatives such as the Nordic Food Policy Lab [80], assembling best practices from two decades of work on food policies and actions in the Nordic countries into a forward-looking solutions menu [18]. The menu covers nutrition, food culture and identity, public food and meals, food waste and sustainable diets, presenting 24 policy examples from local, national and regional levels. Sporting a holistic approach to food policy, they demonstrate how soft policies can deliver important solutions, playing a substantial role in the pursuit of ambitious national and international goals. They have the potential to trigger new conversations, support civil society initiatives like addressing the problems of food waste and inspire new policies in other parts of the world.

### 4.2. Sustainability Action Plans

Including environmental sustainability criteria in updates of current action plans and projects within the field of food and nutrition is a simple and logical way to proceed. The Nordic Plan of Action for Better Health and Quality of Life Through Diet and Physical Activity, which includes visions and goals for the future food and health of the Nordic populations, is a case in point [81]. The Nordic Action Plan has been added to the toolbox for improved diet and health, helping the development of national action plans. Implementing such plans takes resources, so political prioritisation and operative organisation are required.

The Nordic Nutrition Recommendations have also been the basis for other types of collaboration such as the green Keyhole nutrition label. It is used on packaging in Norway, Sweden, Denmark and Iceland (Finland using the essentially equivalent Heart Symbol) to guide consumers to the healthiest choices within each food group according to recommendations for food composition. The symbol has been tested in different groups, showing a positive effect on the health perception of food and behaviour, at least in the short term [82,83]. A similar type of symbol used to guide the population towards more environmentally sustainable choices could be a useful tool of transformation.

Renewed focus on the total food experience in line with the New Nordic Food project, twice funded by the Nordic Council of Ministers, has aimed to lift the joy of food, cooking, Nordic culture and shared experiences. These aspects belong in the transformation tool chest [17].

The close collaboration within food and nutrition between the Nordic countries would have been much harder without the many similarities in food culture and dietary habits. Ultra-processed food items, which have significant environmental impact with little to no health benefits, could easily become one focus area for policy change [84]. Similarities in infrastructures make concerted action easier; collaboration and knowledge sharing enhance the quality of national decisions. The NNR collaboration has contributed essentially to numerous health-related research interventions, campaigns and other smaller and larger projects. The lessons learnt may help to tackle the challenges connected with diet, health and sustainability.

## 5. Conclusions

The Nordic countries, with their tradition of strong political co-operation and communication, could become some of the leaders in making the global food system healthier and more sustainable. The Nordic countries have made strong commitments to adhere to the 2030 UN SDGs and the Nordic Council of Ministers has published a number of reports encouraging a common Nordic policy to obtain more sustainable diets in the Nordic countries.

However, typical current diets in the Nordic countries are neither healthy nor environmentally sustainable, although there are indications of progress towards more fruit and vegetables in the diet. There is a need for both extensive dietary changes and the concurrent transformation of Nordic food production to be able to reach the goals, and an integrated food systems approach, taking into account all steps in agricultural and aquatic production, trade, manufacturing, retailing, consumption and food waste.

Engaging the public to move food consumption closer to the Nordic diet, defined by the future FBDGs of the Nordics focusing on health and sustainability, must happen on many different levels, from individual food and nutrition literacy and skills, societal insight, progressive changes in built food environment to bold policy changes throughout the food system. In order to reach individuals in different target populations, it is important to acknowledge that one message does not fit all.

Official advice to the Nordic population must be evidence-based. An update of the Nordic FBDGs is underway and the EAT-Lancet global reference diet may serve as inspiration. However, an update of the Nordic FBDGs requires updated systematic reviews of the health-based evidence while at the same time integrating environmental sustainability. This comprises an extensive and fast developing scientific area, where updated values on the impact of food products and whole diets on GHG emission, land use, water use, biodiversity, including human and environmental toxicity, are essential. Identifying knowledge gaps is essential and research funding is needed accordingly.

Sustainability aspects should also be incorporated in updates for all current food-related projects such as the Nordic plan of action for nutrition and monitoring as well as national action plans. Monitoring the development of food systems, including health and changes in the Nordic diets through common indicators is important to be able to evaluate initiatives and actions

The Nordic countries are high-income countries, politically based upon an elaborate social model, i.e., the Nordic model, with strong governmental co-operation. They have sufficient financial and political resources to make the dietary changes necessary to decrease the pressure on natural resources and focus on innovative solutions.

Since the Nordic diet can be considered to be a development and improvement of traditional diets, the potential is large for formulating a Nordic diet with gains for both human and planetary health. It is time for concerted engagement and actions—a new Nordic nutrition transition.

## Figures and Tables

**Figure 1 nutrients-11-02248-f001:**
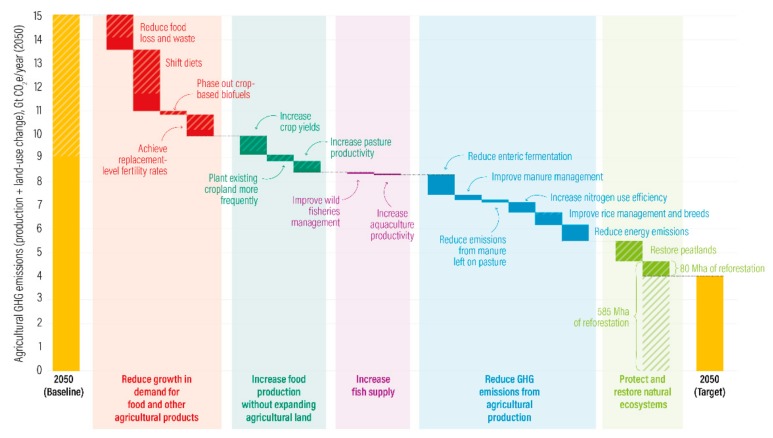
Ambitious efforts across all menu items will be necessary to feed 10 billion people while keeping global temperature increases well below 2 degrees Celsius. Originally printed in ([2]). Reprinted with permission from the World Resources Institute.

**Figure 2 nutrients-11-02248-f002:**
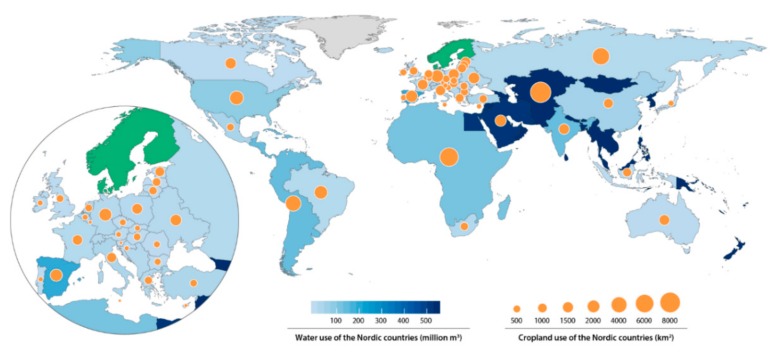
The use of cropland and water outside the Nordic countries as a result of current Nordic food consumption. Originally printed in [36]. Reprinted with permission from Stockholm Resilience Center.

**Figure 3 nutrients-11-02248-f003:**
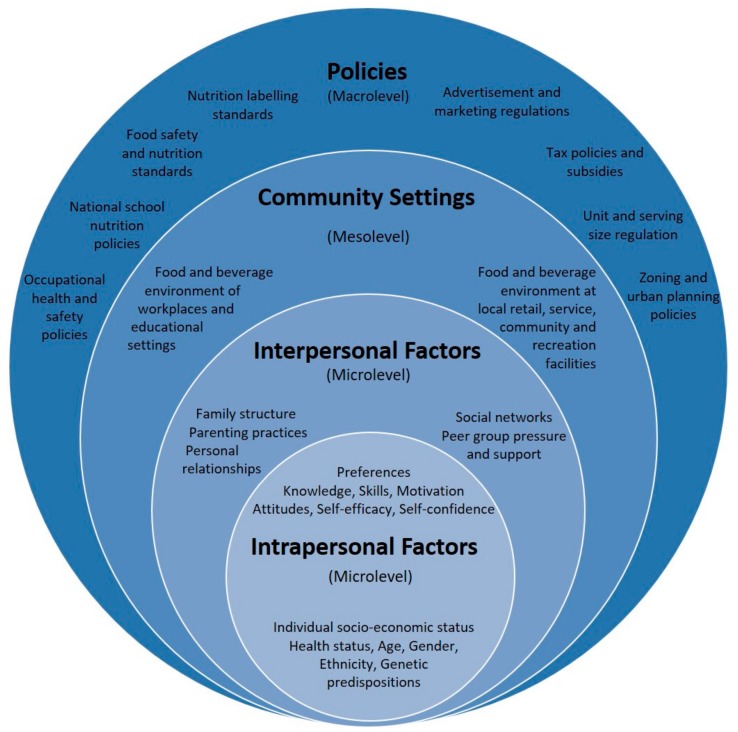
Socio-ecological model of food and beverage intake. Originally printed in [75]. Reprinted with permission from Ludwig-Maximilians-Universität, Munich.

**Table 1 nutrients-11-02248-t001:** The Nordic Food-Based Dietary Guidelines for dietary changes that promote energy balance and health in Nordic populations ^1^.

Increase	Exchange	Limit
Vegetables Pulses	Refined cereals → Wholegrain cereals	Processed meat Red meat
Fruits and berries	Butter → vegetable oils Butter based spreads → Vegetable oil based fat spreads	Beverages and food with added sugar
Fish and seafood	High-fat dairy → Low-fat dairy	Salt
Nuts and seeds		Alcohol

^1^ Originally printed in the Nordic Nutrition Recommendations (NNR)5 [11]. Reprinted with permission from the Nordic Council of Ministers. The colours highlight recommended changes; green for increased intake, red for reduced intake and yellow for changes within a food group.

**Table 2 nutrients-11-02248-t002:** Illustration of foods produced or growing in the wild in the Nordic countries.

Category	Local Foods
Fruits	Apples, pears, plums and cherries
Berries, wild or grown	Bilberries, cowberries, blackcurrants, raspberries, etc.
Vegetables	Cabbage, cauliflower, onion and several leafy vegetables
Roots	Carrots, celery, parsnips, rutabaga, beetroot and potatoes
Herbs	Thyme, parsley, sage, dill, lovage and oregano
Wild plants and mushrooms	Nettles, rosehips and a number of mushroom species
Whole grain	Barley, rye, oats, spelt and buckwheat
Nuts	Hazelnuts, walnuts and chestnuts
Fish and seafood	Sea and lake fish, bivalves and kelp
Meat and eggs	Poultry, beef, lamb, game and birds (farmed or wild and their eggs)

**Table 3 nutrients-11-02248-t003:** Background statistics for the five Nordic countries [8,55].

Metrics	 Denmark	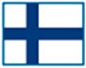 Finland	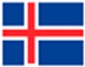 Iceland	 Norway	 Sweden
Population	5,781,190	5,513,130	348,450	5,295,619	10,120,242
Area (km^2^)	42,926	338,430	103,492	323,808	447,435
Total arable land (1000 ha ^1^)	2631	2242	121	986	2568
Percent arable land	61.3	6.6	1.2	3.0	5.7
Arable land per person	0.46	0.41	0.35	0.19	0.25
Arable land, ley and other fodder crops (1000 ha)	488	717	115	483	1082
Sea-fishing, annual catches, tons (2017)	904,476	154,506	1,205,978	2,401,614	222,381
Fish farming, tons of salmon, rainbow trout, cod and halibut	44,380	13,580	15,922	1,286,305	11,361
Organic farmland (% ha, 2017)	9	11	1	5	19
Organic per capita consumption (Euro/person)	278	56	-	80	237
Organic retail sales (Million Euro)	1600	309	-	419	2366
Greenhouse gas (GHG) emissions, tons per person (2015)	9.3	10.2	13.8	10.4	5.5
Recycling of municipal waste (percentage of waste recycled (2016)	28.6	29.2	25.5	28	32.6

^1^ ha; hectare (1 ha equals approximately 2.5 acres).

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
