# Peer review of "Environmental Sustainability Perspectives of the Nordic Diet"

_nutrients, 2019, doi:10.3390/nu11092248_

Round 1

Reviewer 1 Report

The paper is generally well written in a good and clear English.

This paper aims to discuss environmental sustainability aspects of the Nordic diet. Assessment of sustainability aspects relates to elements of the United Nations’ Sustainable Development Goals (SDG).The study presents itself as a review. As the paper does not provide information about review methodology, it is considered as a narrative review rather than a systematic review. Alternatively, one might consider the paper as an academic position paper or a discussion paper.

The paper’s research question appears somewhat vague – rather to discuss than to answer specific research questions. The paper discusses the commonalities between the Nordic countries in terms of political framework, traditions and dietary patterns, and discusses features of the latter in relation to SDG and in relation to climate footprint, water footprint, land use and biodiversity, as well as process aspects like organic food, local food production and food waste reduction. The sustainability assessment of the Nordic diets is conducted as a narrative discussion vis-a-vis individual SDGs rather thanas a formal analysis.

A question may be raised regarding the scientific contribution from the paper. The general message that a recommended diet with less meat, fat and sugar and more plant-based foods also tends to be associated with larger environmental sustainability is not surprising. The paper does not present new research results that document the evidence for this relationship. The paper’s focus on the Nordic countries in this respect may add a twist to the message – but again, the message is not so surprising to an informed reader. I would recommend that the paper is framed as a positionnpaper or commentary rather than a review.

One challenge with the paper is the multidimensionality of sustainability. In particular, it seems a bit awkward to consider process aspects like organic and local production and food waste reduction on an equal footing with GHG emissions, water footprint, land occupation and biodiversity preservation. This is awkward, because the former three tend to be instruments rather than end goals. From a sustainability point of view, for example, food waste is a problem because it implies ”unnecessary” footprints on climate, water, land and biodiversity.

The paper adopts the view that local production is more sustainable. Whereas it is evident that local production tends to require less transportation, efficient ressource utilization in food production is crucial. If local production utilizes resources less efficiently than production abroad - e.g. due to less favorable climatic conditions which could well be the case for crop production in Northern Europe - local production may well be less sustainable than consumption of imported products.

Reviewer 2 Report

The authors are to be commended on their paper. It is encouraging to read about countries collaborating with the intention of encouraging and supporting both healthy and sustainable dietary behaviours. 

This manuscript provides a unique contribution to the field, demonstrating the complexities and influencing factors on behaviour change. The identified areas for action and collaboration have a practical application, which will hopefully progress.

Specific dietary behaviours relating to a healthy and sustainable diet were discussed, such as locally produced food, organic farming and food waste. However, there was little emphasis placed on the impact of ultra processed energy-dense nutrient-poor foods (often packaged), which have a significant environmental impact with little to no health benefits. If the intake of these foods is high in the Nordic countries, as is the case in other high income countries, I believe this is worth including as a focus area for policy change. 

Evidence relating to the health benefits, or lack of, associated with the intake of organic foods could have been strengthened. There were comments made on lines 196-197 on the influence of social class on dietary intake, in addition to comments regarding the higher cost of organic food for the consumer. A brief discussion on the health and environmental benefits of increasing plant-based foods (regardless of whether grown organically or not) would help address issues related to socio-economic barriers. 

In section 4.2, there was suggestion of a potential food labelling system to help consumers identify the more environmentally sustainable choices. This is a very valid suggestion and I believe a simple front-of-pack labelling icon identifying the GHG emissions of specific foods has been trialled in the United Kingdom. 

This manuscript provides a holistic view of the issues when addressing both health and environmentally sustainability in dietary recommendations. The situation in the Nordic countries appears unique and this manuscript demonstrates that by collaborating these countries could lead the way in this field. Other middle and high income countries could monitor and learn.

There were grammatical errors throughout the manuscript, with a few mentioned below. 

Inconsistencies between 'Nordic Diet' and 'Nordic diet'

Please review colloquial terms used, for example in lines 94, 391 and 107.

Line 26: Change plant-based food to 'foods'

Line 41: Please remove 'e.g.' prior to the in text citations

Line 181: Change 'are still' to 'is still'

Line 189: Should read '.... and the prevalence of overweight and obesity.'

Line 217: NCDs have been previously abbreviated earlier in the manuscript

Line 222: Insert 'and' between Denmark and Finland

Line 233: GHG emissions previously abbreviated in line 82

Line 285: Insert comma between 'diets' and 'respectively'

Line 289: In-text citation for Brink et al 2019 needs to be reformatted

Line 297: Insert space between 'whichidentified'

Lines 371 - 377: Please change to 'Firstly' 'Secondly' and 'Lastly' as this is grammatically correct

Line 386: Change 'aims' to 'aim'

Line 417: Specify if the food bought and thrown away is based on households/individuals?

Line 442: Change 'other' to 'others'

Reviewer 3 Report

This reviewer finds this paper to be sound: it includes most recent findings and compelling arguments to substantiate the shifts being proposed.

A few minor (and I mean minor) corrections have been inserted as recommendations into the paper, see attached.'

Round 2

Reviewer 1 Report

The re-labelling of the paper to a 'concept paper' has solved most of my previous issues.

Even though I still think it is a bot awkward to consider local or organic production and reduced food waste as sustainability dimensions per se, I will not block publication of the paper for tis reason.

Author Response

Oslo September 12, 2019

Dear Editor,

Thank you for the decision to accept after minor revision. We have addressed your comments as follows.

Comments:

The authors have revised the manuscript following the reviewers’ suggestions and comments to a high degree with a nice result. With the approach taken in this manuscript to provide an overview of the current situation in relation to the concept of environmental sustainability perspectives of the Nordic Diet in a concept paper, it seems odd that the authors in section 2.3 on ”The Nordic diet and health” do not include the biggest intervention study run on Nordic Diets, namely the OPUS study. I recommend that the authors consider to include the two major papers from this study. 1. Poulsen SK, Due A, Jordy AB, Kiens B, Stark KD, Stender S, Holst C, Astrup A, Larsen TM. Health effect of the New Nordic Diet in adults with increased waist circumference: a 6-mo randomized controlled trial. Am J Clin Nutr. 2014 Jan;99(1):35-45. doi: 10.3945/ajcn.113.069393. Epub 2013 Nov 20. PMID: 24257725 where the main conclusion was that an ad libitum NND produces weight loss and blood pressure reduction in centrally obese individuals. 2. Damsgaard CT, Dalskov SM, Laursen RP, Ritz C, Hjorth MF, Lauritzen L, Sørensen LB, Petersen RA, Andersen MR, Stender S, Andersen R, Tetens I, Mølgaard C, Astrup A, Michaelsen KF. Provision of healthy school meals does not affect the metabolic syndrome score in 8-11-year-old children, but reduces cardiometabolic risk markers despite increasing waist circumference. Br J Nutr. 2014 Dec 14;112(11):1826-36. doi: 10.1017/S0007114514003043. Epub 2014 Oct 17. This paper concludes that NND school meals did not affect the MetS score in healthy 8-11-year-olds (as small improvements in blood pressure, TAG concentrations and insulin resistance were counterbalanced by slight undesired effects on waist circumference and HDL-cholesterol concentrations).

Answer: We fully agree that the OPUS study needs to be cited, and a new sentence has been added in section 2.3 on to “The Nordic diet and health” saying “One example is the OPUS study, a crossover intervention NND school meal trial in 834 Danish children aged 8-11 years [19]” Reference 19 (new) is Andersen et al. Effects of school meals based on the New Nordic Diet on intake of signature foods: a randomised controlled trial. The OPUS School Meal Study. The British Journal of Nutrition 2015.

Furthermore, both of the papers you suggested to include has been included, Poulsen et al. is now reference 20 and Damsgaard et al. is now reference 21.

It is recommended that the word ’better’ in line 447 is replaced with higher; and that ’old’ in line 627 is deleted.

Answer: The word ‘better’ has been replaced with ‘higher’ (now line 405) and the word ‘old’ has been deleted (now line 573).

Comment in the e-mail: The Academic Editors recommend that your manuscript should undergo extensive English editing. We suggest that you have your manuscript checked by a native English speaking colleague or use a professional English editing service.

Answer: Our manuscript has been edited by a native English speaker in the communication department of our Institute and a number of corrections have been implemented (highlighted by track changes).

Please check that our graphical abstract is not the one first uploaded, but a slightly revised version sent by email to Jill Sun some days later.
